# Characterization and Engineering of Two Novel Strand-Displacing B Family DNA Polymerases from *Bacillus* Phage SRT01hs and BeachBum

**DOI:** 10.3390/biom15081126

**Published:** 2025-08-05

**Authors:** Yaping Sun, Kang Fu, Wu Lin, Jie Gao, Xianhui Zhao, Yun He, Hui Tian

**Affiliations:** Research Center of Molecular Diagnostics and Sequencing, Research Institute of Tsinghua University in Shenzhen, Shenzhen 518000, China; sunyp@tsinghua-sz.org (Y.S.);

**Keywords:** Phi29-like processivity, Mg^2+^-dependence modified substrate incorporation, exonuclease activity

## Abstract

Polymerase-coupled nanopore sequencing requires DNA polymerases with strong strand displacement activity and high processivity to sustain continuous signal generation. In this study, we characterized two novel B family DNA polymerases, SRHS and BBum, isolated from *Bacillus* phages SRT01hs and BeachBum, respectively. Both enzymes exhibited robust strand displacement, 3′→5′ exonuclease activity, and maintained processivity under diverse reaction conditions, including across a broad temperature range (10–45 °C) and in the presence of multiple divalent metal cofactors (Mg^2+^, Mn^2+^, Fe^2+^), comparable to the well-characterized Phi29 polymerase. Through biochemical analysis of mutants designed using AlphaFold3-predicted structural models, we identified key residues (G96, M97, D486 in SRHS; S97, M98, A493 in BBum) that modulated exonuclease activity, substrate specificity and metal ion utilization. Engineered variants SRHS_F and BBum_Pro_L efficiently incorporated unnatural nucleotides in the presence of Mg^2+^—a function not observed in Phi29 and other wild-type strand-displacing B family polymerases. These combined biochemical features highlight SRHS and BBum as promising enzymatic scaffolds for nanopore-based long-read sequencing platforms.

## 1. Introduction

Bacteriophage-derived B family DNA polymerases with strand displacement activity, including Phi29, T4, IME199, Bam35, and RB69, have been extensively studied due to their unique biochemical properties and broad applications in molecular biology [1,2,3,4,5]. Phi29 DNA polymerase (phi29 DNAP) stands out for its exceptional strand displacement activity, high processivity, and high fidelity. These properties make it indispensable for multiple displacement amplification (MDA), isothermal DNA amplification, whole-genome amplification, and next-generation long-read sequencing techniques [6,7,8]. The ability to unwind and displace downstream DNA strands without dissociating from the template—a hallmark of strand-displacing polymerases—enables continuous elongation that is particularly advantageous for long-read sequencing technologies, such as polymerase-driven single-molecule nanopore sequencing. However, this critical feature is not uniformly conserved among B family polymerases. While enzymes like IME199 and Bam35 DNA polymerases exhibited robust strand displacement activity suitable for rolling circle amplification (RCA) [3,5], others such as T4 DNA polymerase lacked this capability entirely. Consequently, the discovery and engineering of novel B family DNA polymerases with robust strand displacement activity remain important research priorities.

DNA polymerases require divalent metal ions to catalyze the nucleotidyl transfer reaction. Magnesium (Mg^2+^) is the predominant cofactor under physiological conditions, ensuring high replication fidelity by stabilizing dNTP conformations and promoting correct base pairing recognition. However, alternative metal ions, including manganese (Mn^2+^) [9], iron (Fe^2+^) [10], cobalt (Co^2+^) [11,12,13], calcium (Ca^2+^) [14,15], cadmium (Cd^2+^) [13,16], copper (Cu^2+^) [16], and zinc (Zn^2+^) [12], can also support catalysis in some polymerases under specific conditions. Ion specificity of DNA polymerases depends on the ions’ chemical properties, coordination geometry, and their interactions with the enzyme’s active site [9,17,18]. They will affect the processivity, substrate compatibility and fidelity of DNA polymerases. For example, Mn^2+^ was particularly effective in enabling polymerases to incorporate modified or unnatural nucleotide analogs due to its flexible coordination geometry and larger ionic radius [17,19]. Further, Mn^2+^ for Bst exo- polymerases has been shown as the most effective alternative cofactor to prevent multimerization, which is non-specifical DNA synthesis [16,20]. However, its use severely compromises fidelity, potentially leading to replication errors and genomic instability [21]. For applications like polymerase-driven single-molecule nanopore sequencing, which utilizes polymer-tagged nucleotides to generate base-specific signals [22], polymerases are required to efficiently incorporate modified substrates without sacrificing Mg^2+^-dependent fidelity. In this study, we sought to identify and engineer DNA polymerases with this specific capability.

We presented the biochemical characterization of two novel B family DNA polymerases—SRHS and BBum—from *Bacillus* phages SRT01hs and BeachBum. Both enzymes exhibited robust strand displacement activity, 3′→5′ exonuclease proofreading, and 5′→3′ synthetic activity across a physiologically relevant temperature range (10–45 °C) and multiple catalytic ion conditions (Mg^2+^, Mn^2+^, and Fe^2+^). Leveraging AlphaFold3 for structural prediction and Phi29-based comparative modeling, we identified conserved catalytic motifs and unique structural features that guided our rational engineering approach.

We engineered SRHS and BBum variants by targeting exonuclease, catalytic, and substrate-binding domains using structural alignment, electrostatic analysis, and insights from Phi29-related enzymes. We assessed these variants for exonuclease activity, processivity and Mn^2+^/Mg^2+^-dependent incorporation of polymer-tagged unnatural nucleotides.

Our findings demonstrate mutations N63D (BBum_Exo_B) and Y166C (BBum_Exo_C) preserved exonuclease activity—contrasting with Phi29 equivalents N62D and Y165C [23]. Interestingly, several engineered variants, particularly SRHS_F and BBum_Pro_L, achieved efficient Mg^2+^-dependent synthesis using unnatural substrates, a capability absent in wild-type strand-displacing B family polymerases. Critical residues (G96R/M97K/D486E in SRHS; S97R/M98K/A493E in BBum) were identified for substrate compatibility. These results significantly advance our understanding of polymerase engineering and highlight the potential of SRHS and BBum as next-generation enzymes for applications like nanopore sequencing.

## 2. Materials and Methods

### 2.1. Materials

DNA polymerase SRHS and DNA polymerase BBum were derived from *Bacillus* phage SRT01hs and *Bacillus* phage BeachBum, respectively. Wild-type gene sequences for SRHS (Accession No: YP_009910648.1), BBum (Accession No: YP_009910184.1) and Phi29 (Accession No: P03680.1.) were obtained from the NCBI database.

### 2.2. Plasmid Construction and Protein Expression

The coding sequences of Phi29 polymerase, wild-type SRHS and BBum polymerases, and other mutant variants were codon-optimized for *E. coli* expression, synthesized and cloned into the Pet30a plasmids by Genscript Biotechnology (Nanjing, China). All constructs included C-terminal TEV-cleavable His-tags.

### 2.3. Protein Expression

All protein expression plasmids were transformed into chemically competent BL21 (DE3) *E. coli* cells (ToloBio, Wuxi, China) according to the manufacturer’s instructions. Briefly, BL21 (DE3) competent cells were thawed on ice for 2 min before adding 100 ng of plasmid. After a thirty-minute incubation on ice, cells underwent heat shock at 42 °C for 90 sec using a precision metal bath (H203-100C, COYOTE, Beijing, China), and were then placed on ice for 3 min. A total of 500 μL of LB medium without an antibiotic was added to the cells to recover on a shaker (200 rpm) at 37 °C for 45 min. After centrifugation, the cells were resuspended in 200 μL of LB medium without an antibiotic and were cultured on a 10 cm LB agar plate containing kanamycin overnight.

Single colonies were cultured in 5 mL of LB medium supplemented with kanamycin (50 µg/mL) overnight at 37 °C. This culture was further incubated in 200 mL of liquid LB medium at 37 °C on a shaker at 200 rpm until an OD_600_ value of 0.6 was achieved. Protein expression was induced by adding isopropyl-D-1-thiogalactopyranoside (IPTG) up to a concentration of 1 mM (A100487, Sango Biotech, Shanghai, China), followed by incubation at 16 °C for 14–16 h. The cells were centrifuged at 8000 rpm for 10 min, collected, washed by ice-cold PBS twice, and then stored at −80 °C until further use.

All protein purification steps were performed at 4 °C. The cells were resuspended in 50 mL of binding buffer (50 mM Tris-HCl [pH 7.5] and 500 mM NaCl) and then lysed by sonication (100 W, 2 s on bursts, and 5 s intervals) using a Scientz-IID sonicator (01C1503, Scientz Biotechnology, Ningbo, China) on ice for 20 min. After centrifugation at 12,000 rpm for 30 min, the soluble protein was collected. The expressed His-tagged target proteins were purified using an in-house Ni Sepharose 6FF gravity column. Namely, 1.3 g of cells from a 200 mL bacterial solution were collected, and 3 mL of Ni Sepharose 6FF resin (17531801, Cytiva, Marlborough, MA, USA) (50% suspension) was used for purification. The resin was packed in a 10 mL gravity chromatography column (7321010, Bio-Rad, Hercules, CA, USA), rinsed with 5 column volumes of double-distilled water, and equilibrated with 10 column volumes of binding buffer (50 mM Tris-HCl [pH 7.5] and 500 mM NaCl). Protein supernatants were slowly added to the chromatography column. This loading process was repeated three times. The column was then washed with 10 column volumes of binding buffer before eluting the target protein with an imidazole step gradient (20–300 mM) in elution buffer (20 mM Tris, 500 mM NaCl).

The target proteins were detected via homemade 10% SDS-PAGE. High-purity elution components then underwent 10 K ultrafiltration (157655, Millipore, Billerica, MA, USA) to remove imidazole and obtain a high-concentration protein. To remove His-tags, 400 μL of TEV (300 ng/μL) enzyme was used to digest 600 μg of target protein overnight at 4 °C in TEV buffer (50 mM Tris-HCl, 5 mM NaCl, 1 mM EDTA, 1 mM DTT, pH 8.0). Due to the TEV protease with His-tags, an in-house Ni Sepharose 6FF gravity column was used to remove the TEV protease and the cleaved His-tags in the reaction mixture. The target proteins were detained in the flow-through. The target proteins were further detected via 10% SDS-PAGE and underwent 10 K ultrafiltration to obtain a high-concentration proteins. The concentrations of proteins were calculated by the Pierce BCA Protein Assay Kit (23225, Thermo Fisher Scientific, Waltham, MA, USA). The protein purity was tested by homemade SDS-PAGE. Protein concentrations were further checked with reference standard of bovine serum albumin (BSA).

Homemade SDS-PAGE was enabled by the 10% PAGE gel quick preparation kit. For protein electrophoresis, 10 μL of protein samples and 2.5 μL of 6 × protein loading buffer were mixed, then heated at 95 °C for 5 min and finally centrifuged at 12,000 rpm for 1 min. The buffer used for electrophoresis was tris-glycine buffer. The electrophoresis condition was 200 V for 35 min. The eStain L1 (L00657C, Genscript, Nanjing, China) was used to stain the protein gels.

### 2.4. Structural Analysis

Protein structures were predicted using AlphaFold3 for the SRHS and BeachBum polymerases [24], while the Phi29 polymerase reference structure (PDB ID: 2PYL) was obtained from the Protein Data Bank (PDB) [25]. The structure visualization and electrostatic potential maps of the surface were created and analyzed using PyMOL (v 2.3, Schrödinger Inc., New York, NY, USA) [26].

### 2.5. Exonuclease Activity

To evaluate exonuclease activity, we designed four oligonucleotide substrates: M25 (unmodified, TCCTAACGAGATTAGTTTTGCTGTT), M25-3′ (with three consecutive phosphorothioates (PSs) at the 3′ terminal, TCCTAACGAGATTAGTTTTGCT*G*T*T), M25-5′ (with three consecutive PSs at the 5′ terminal, T*C*C*TAACGAGATTAGTTTTGCTGTT), and M25-3′5′ (with PSs at both the 3′ and 5′ terminals, T*C*C*TAACGAGATTAGTTTTGCT*G*T*T). Three constitutive PSs as a protecting group can protect oligonucleotides from degrading.

The exonuclease activities of DNA polymerases were tested using the four specific oligonucleotides. Briefly, reactions containing 1 μM oligonucleotides and 200 nM polymerase in phi29 buffer (B0269, NEB, Ipswich, MA, USA) were incubated at 30 °C for 30 min. Products were analyzed by 10% TBE-urea PAGE (EC68752BOX, Thermo Fisher Scientific, Waltham, MA, USA).

### 2.6. The Preparation of Single-Strand Circular (SSC) Templates

The single-stranded DNA template (200bp, AGGTCGCCAGTGAAGTCTTTCGGGCTTCCTCTTAATCTTTTTGATGCAATCCGCTTTGCTTCTGACTATAATAGTCAGGGTAAAGACCTGATTTTTGATTTATGGTCATTCTCGTTTTCTGAACTGTTTAAAGCATTTGAGGGGGATTCAATGAATATTTATACCGATTCCGCAGTATTGCACTCTATCGTCGCCAGCCC) and a primer (CTGGCGACCTGGGCTGGCGAC) were synthesized by Genscript Biotechnology (Nanjing, China). A 100 nM single-stranded DNA template was mixed with a 100 nM primer in 1× T4 DNA ligase buffer (NEB, Ipswich, MA, USA), followed by a stepwise annealing process (95 °C to 25 °C). Circularization was performed with T4 DNA ligase (400 U/reaction, NEB, Ipswich, MA, USA) at 16 °C overnight, followed by heat inactivation at 65 °C for 10 min. To remove residual linear DNA, reactions were treated with Exonuclease I (10 U/reaction, NEB, Ipswich, MA, USA) and Exonuclease III (100 U/reaction, NEB, Ipswich, MA, USA) in Exo I buffer at 37 °C for 1 h, then heat-inactivated at 80 °C for 15 min. The single-stranded circular DNA products were purified using the Zymo Oligo Clean & Concentrator Kit (Zymo Research, Irvine, CA, USA) according to the manufacturer’s protocol, quantified using the Qubit ssDNA Assay Kit (Thermo Fisher Scientific, Waltham, MA, USA), and verified by 6% TBE-Urea denaturing PAGE.

### 2.7. Rolling Circle Amplification (RCA)

The template–primer complex was prepared by annealing 800 nM of a phosphorothioate-modified primer (5′-CTGGCGACCTGGGCTGGC*G*A*C-3′) and 200 nM of SSC templates in an annealing buffer (20mM HEPES, pH = 7.5). The mixture was heated to 95 °C for 3 min followed by gradual cooling to room temperature over 25 min.

To explore processivity of polymerases under varying temperatures and ion conditions, reactions were prepared using 100 nM of a template–primer complex, comprising a 100 nM template and 400 nM primer, 100 nM polymerase, and 100 μM dNTPs. For temperature-dependent assays, reactions were carried out in reaction buffer 1 (1 mM MgCl_2_, 150 mM KCl, 20 mM HEPES, 4 mM TCEP, pH 7.5) at five different temperatures: 10 °C, 25 °C, 35 °C, 45 °C, and 55 °C. For ion-dependent assays, reactions were conducted in reaction buffer 2 (150 mM KCl, 20 mM HEPES, 4 mM TCEP, pH 7.5) supplemented with one of five divalent cations: 1 mM MgCl_2_, MnCl_2_, CaCl_2_, SrCl_2_, or FeSO_4_. The reactions were performed for 60 min and then terminated by adding 0.5 M EDTA. To prevent the oxidation of Fe^2+^ to Fe^3+^ during RCA, all Fe^2+^-containing reactions were conducted under nitrogen-purged conditions. Specifically, FeSO_4_ stock solutions were freshly prepared in deoxygenated water and kept on ice. Reaction buffers and enzyme mixtures were purged with ultrapure nitrogen gas for at least 30 min prior to use. Fe^2+^ was then added immediately before initiating the reactions. Each assay was repeated thrice for every polymerase available.

To explore processivity of polymerase mutant variants using dNTPs, 100 nM of the template–primer complex was incubated with 100 nM of polymerase variants and 100 μM of natural substrates in reaction buffer 1. The reactions were performed at 30 °C for 60 min.

The four kinds of unnatural substrates were dA6P-Cy3-dT_4_-FldT-dT-FldT-dT23-C3, dT6P-Cy3-dT_2_-dSp_8_-dT_20_-C3, dC6P-Cy3-dT_4_-dSp_8_-dT_23_-C3, and dG6P-Cy3-T_30_-C3. They were synthesized according to previous research [27].

To explore the processivity of polymerases using unnatural substrates under various ions conditions, 100 nM of the template–primer complex was mixed with 100 nM of polymerase proteins and 100 μM of unnatural substrate in reaction buffer 2 under the following conditions: 1mM MnCl_2_, 1 mM MgCl_2_, 1 mM CaCl_2_, 1 mM SrCl_2_, and 1 mM FeSO_4_.

To explore the processivity of polymerase mutant variants with unnatural substrates in the presence of Mn^2+^ or Mg^2+^, 100 nM of the template–primer complex was incubated with 100 nM of each polymerase variant, 1 mM MnCl_2_ or 1 mM MgCl_2_, and 100 μM unnatural substrate in reaction buffer 2. The reactions were performed for 60 min.

The RCA products were detected by 0.6% agarose gel electrophoresis. The GeneRuler High Range DNA Ladder and 1 Kb Plus DNA Ladder worked as DNA molecular weight standards. Following 2–3 h of electrophoresis, SYBR Gold-stained gels were imaged using AzureSpot software (version 2.1; Azure Biosystems, Dublin, CA, USA).

### 2.8. DNA Replication Using a Hairpin Linear Template and Unnatural Substrates

A linear template was used to explore the ion selectivity of various polymerases. The sequence of the template was a hairpin structure which contained a 3′ self-priming region, allowing for primer-independent initiation of DNA synthesis (TTTTTGCGCTCGAGATCTCCGTAAGGAGATCTCGAGCGCGGGACTACTACTGGGATCATCACTGCCACCTCAGCTGCACGTAAGTGCAGCTGAGGT*G*G*C). The template was denatured at 95 °C for 3 min in an annealing buffer (20 mM HEPES, pH 7.5) and gradually cooled to room temperature over 25 min to facilitate proper secondary structure formation.

To explore replication ability of polymerase mutant variants using unnatural substrates, 20 nM of the hairpin template was incubated with 10 nM of polymerase variants and 100 μM of unnatural substrates in reaction buffer 3 (0.4 M potassium acetate, 0.05 M Tris-HCl, 20 mM MgOAc, and 4 mM TCEP). The reactions were performed for 30 min. The results were analyzed by using 10% TBE-Urea gel in an XCell SureLock Mini-Cell (EI0001, Thermo Fisher Scientific, Waltham, MA, USA), as recommended by the manufacturer. All electrophoresis was performed at 180 V for 45 min and visualized by SYBR Gold staining (S11494, Invitrogen, Life Technologies, Carlsbad, CA, USA). The O’RangeRuler 10 bp DNA Ladder was used as the DNA molecular weight standard.

### 2.9. Polymerase Fidelity Analysis

A comprehensive protocol for fidelity assessment is provided in Appendix A.

## 3. Results

### 3.1. SRHS and BBum DNA Polymerases Exhibit Phi29-like Strand Displacement, Exonuclease Activity, and Processivity

Phi29 DNA polymerase, a prototypical strand-displacing B-family polymerase, contains several conserved structural domains, including an N-terminal 3′→5′ exonuclease domain responsible for proofreading, a palm domain that harbors the catalytic residues for nucleotide incorporation, a fingers domain that guides the incoming dNTPs and stabilizes the template strand, and a thumb domain involved in DNA binding and maintaining processivity [28,29]. In addition, it has two tetratricopeptide repeat (TPR) subdomains, TPR1 and TPR2. TPR1 is primarily involved in template binding and correct positioning of the primer terminus, whereas TPR2 plays a crucial role in strand displacement by stabilizing the displaced strand and facilitating efficient translocation along the template [29,30]. These domain features collectively account for the high fidelity, robust strand displacement activity, and exceptional processivity of DNA polymerases. To predict the functions of the SRHS and BBum DNA polymerases isolated from *Bacillus* phages SRT01hs and BeachBum, respectively, we employed AlphaFold3 to predict their tertiary structures. Both predicted models exhibited exceptionally high confidence (pTM = 0.95; pLDDT mean > 90), exceeding the established threshold for reliable topology prediction (pTM > 0.8) [31]. Despite sharing less than 75% sequence identity with Phi29 (55.92% between BBum and Phi29; 54.04% between SRHS and Phi29), the predicted structures revealed high structural similarity (Figure 1A), confirming their classification within strand-displacing B family DNA polymerases. This structural conservation suggested that functional features such as 3′→5′ exonuclease proofreading, strand displacement, and high processivity may be retained.

We characterized the exonuclease activities of the SRHS and BBum polymerases using four oligonucleotide substrates: an unmodified control (M25) and 3′-protected (M25-3′ with three consecutive PSs at the 3′ terminal), 5′-protected (M25-5′ with three consecutive PSs at the 5′ terminal), and doubly protected (M25-3′5′ with three consecutive PSs at both terminals) variants. PS modifications have been known to effectively inhibit exonuclease activity, as reported for VpV62, PB polymerases, and some psychrophilic polymerases [32,33,34,35]. Following protein purification via His tags, protein concentration and purity were determined using the Pierce BCA Protein Assay Kit and SDS-PAGE (Appendix A). All three polymerases showed complete degradation of the unmodified and 5′-protected oligonucleotides, while largely preserving the 3′-protected and doubly protected oligonucleotides (Figure 1B and Appendix A). These results confirmed that SRHS and BBum polymerases possessed robust 3′→5′ exonuclease activity with negligible 5′→3′ exonuclease activity.

Rolling circle amplification (RCA) using a primed single-strand circular (SSC) template (200 bp) revealed that the SRHS and BBum polymerases possessed strong strand displacement and high processivity comparable to Phi29 (Figure 1C), as evidenced by production of high-molecular-weight DNA products (>48 kb).

Temperature-dependent activity analysis demonstrated polymerase activity between 10 and 45 °C, with optimal performance at 35 °C. Only residual activity was observed at 55 °C, indicating functional inactivation at this upper temperature (Figure 2A–C).

DNA polymerases required divalent metal ions as catalytic cofactors to facilitate extension. We assessed the metal cofactor selection by RCA with various metal ions (Mg^2+^, Mn^2+^, Fe^2+^, Ca^2+^ and Sr^2+^). These metal ions possess distinct ionic radii: Mg^2+^ (0.72 Å), Mn^2+^ (0.83 Å), Fe^2+^ (0.78 Å), Ca^2+^ (1.00 Å), and Sr^2+^ (1.18 Å) [36]. Mg^2+^ and Mn^2+^ serve as primary catalytic cofactors for DNA polymerases due to their optimal size for precise transition-state stabilization in phosphoryl transfer reactions. Ca^2+^ and Sr^2+^—with significantly larger radii—act as negative controls, as their geometric mismatch disrupts active-site coordination and prevents catalysis. Both ions are often used in nanopore sequencing systems to produce stagnation signals [22,37]. Although Fe^2+^ has an ionic radius comparable to that of Mg^2+^ and Mn^2+^, its potential role as a catalytic cofactor for DNA polymerase remains unclear. RCA results showed functional activity with Mg^2+^, Mn^2+^, and Fe^2+^, but not with Ca^2+^ or Sr^2+^ (Figure 2D).

In addition, urea tolerance was assessed using RCA. As is shown in Appendix A, urea inhibited extension efficiency of DNA polymerases in a concentration-dependent manner. Notably, at 1 M urea, BBum retained a certain level of amplification activity, outperforming both SRHS and Phi29 under the same conditions.

Collectively, these findings establish SRHS and BBum as Phi29-like polymerases with robust strand displacement capability, 3′→5′ exonuclease activity, and high processivity across a broad range of physiologically relevant conditions.

### 3.2. Rational Design of SRHS and BBum Polymerase Mutants

To elucidate the molecular determinants of exonuclease activity and processivity, we performed structure-guided mutagenesis based on AlphaFold3 predictions and comparative analysis with Phi29 polymerases.

#### 3.2.1. Mutants Targeting Exonuclease Activity

Structural analysis of the SRHS polymerase identified a distinctive salt-bridge interaction between Lys129 and Asp538 that differed from corresponding residues in the Phi29 and BBum polymerases (Figure 3A). This interaction network, potentially stabilized by adjacent Gln132 through polar contacts, may influence both exonuclease activity and strand displacement ability. To investigate these structural features, we generated a series of SRHS variants:

SRHS_A (K129S): Mutation of Lys129 to serine, located near the exonuclease site, was hypothesized to impact exonuclease activity by disrupting key interactions.

SRHS_B (K129S and D538A): This double mutation further targeted Asp538, which, in combination with K129S, was expected to modify the exonuclease activity by weakening the critical interactions in the exonuclease active site.

SRHS_C (K129S, Q132K, and D538K): This triple mutation altered both Lys129 and Gln132, along with Asp538, to evaluate the synergistic effects of these changes on exonuclease activity.

Further examination of the template-binding region revealed Ser187 and Asn196 as potential mediators of strand displacement ability (Figure 3A and Appendix A). We hypothesize that mutating these residues to Arg187 and Lys196, respectively, could significantly enhance the stabilization of template binding to improve the processivity and enable polymerase extension under adverse conditions. To validate the hypothesis, we designed the SRHS_D (K129S, Q132K, S187R, N196K, and D538K) variant to test whether introducing basic residues at positions 187 and 196 could enhance template binding while maintaining modified exonuclease activity, potentially improving polymerase activity.

In addition, D12 and D169 of the Phi29 DNA polymerases interacted directly with single-stranded DNA (ssDNA) to create catalytic carboxylates in the exonuclease domain [2,23,38] (Figure 3A). Mutation of both residues to alanine lowered exonucleolytic activity by 10^5^ [29]. N62D of Phi29 DNA polymerases directly disturbed the stable primer strand binding to reduce the exonuclease activity [39]. Mutation of its Y165C also negatively affects catalysis of exonuclease activity without significantly changing the affinity of polymerase for ssDNA [40]. We speculated that these key exonuclease residues might be highly conserved across strand-displacing B family DNA polymerases and thus may exert similar effects on the exonuclease activity of both SRHS and BBum polymerases. To test this hypothesis, we used BBum polymerase as a representative and introduced alanine, aspartate, cysteine, and alanine substitutions at positions Asp12, Asn63, Tyr166, and Asp170, respectively, generating the BBum variants (Figure 3A).

BBum_Exo_A (D12A): Mutation of Asp12 to Ala, located in the center of the exonuclease activity site, was hypothesized to impact exonuclease activity by disrupting the interaction with ssDNA.

BBum_Exo_B (N63D), BBum_Exo_C (Y166C) and BBum_Exo_D (D170A): Mutation of Asn63 to Asp, Tyr166 to Cys, and Asp170 to Ala, located at the edge of the exonuclease activity center, was hypothesized to impact the exonuclease activity, similar to N62D, Y165C, and D169A in Phi29 DNA polymerases.

#### 3.2.2. Mutagenesis Targeting Processivity and Strand Displacement

Building upon previous demonstrations of Phi29 polymerase’s ability to incorporate modified nucleotides under Mn^2+^ conditions [27], we investigated whether the SRHS and BBum polymerases possessed similar capabilities through targeted mutagenesis of key functional regions. Structural analysis of the catalytic center and primer-binding region identified critical residues for regulating substrate recognition and extension: Gly96, Met97, Asp486, and Lys515 in the SRHS polymerases and Ser97, Met98, and Ala493 in BBum (Figure 3A and Appendix A).

For the SRHS polymerases, we created two comprehensive variants: SRHS_E (M97K, K129S, D486E, K515Y, D538A) and SRHS_F (G96R, M97K, K129S, D486E, K515Y, D538A). These variants were designed to modify residues of the catalytic center (Asp486, Lys515) and primer-binding region (Gly96, Met97) to assess their effect on processivity and strand displacement. Both constructs also incorporated previously identified mutations in the exonuclease domain to assess combined effects on polymerase performance.

For BBum polymerases, nine mutant variants were designed to target these residues in the catalytic center and primer binding region:

BBum_Pro_E (D150H, A151K, P152E): Compared with Phi29 polymerases, BBum polymerases showed a narrow and acidic entrance around the exonuclease activity center. We mutated Asp150, Ala151, and Pro152 to His, Lys and Glu according to the sequence alignment to alter electrostatic properties and potentially modify the exonuclease activity center (Figure 3B and Appendix A).

BBum_Pro_F (A493E), BBum_Pro_G (S97R, M98K), and BBum_Pro_H (S97R, M98K, A493E): These mutants targeted catalytic and primer-binding residues (Ser97, Met98, Ala493) to investigate their impact on processivity and strand displacement.

Sequence alignment and comparative structural analysis between BBum and Phi29 polymerases identified a distinctive protruding structure domain with significant length variation near the substrate entrance (Figure 3B,C). To investigate structure–function relationships, we generated the following chimeric constructs.

BBum_Pro_I and BBum_Pro_J replaced BBum residues 510–542 and 516–542, which formed a protruding structure near the substrate entrance, respectively, with corresponding Phi29 sequences (residues 501–524 and 507–524), while maintaining the S97R/M98K/A493E background. Further refinement produced BBum_Pro_K and BBum_Pro_L, incorporating an additional K512Y mutation within the Phi29-derived regions. All of them were designed to evaluate their effect on polymerase processivity and strand displacement under various conditions.

In addition, we generated BBum_Pro_M by replacing residues 510–542 with a shortened segment from *Geobacillus* phage CPS2 polymerase (UniProt ID: A0A0K2P6L7), testing the functional impact of a more compact substrate-entry structure (Figure 3B).

All SRHS and BBum mutant variants are summarized in Table 1.

### 3.3. Exonuclease Activity in SRHS and BBum Mutants

Both the SRHS and BBum DNA polymerases possessed intrinsic 3′→5′ exonuclease activity. To evaluate how specific mutations affected this activity, we incubated M25-5′ oligonucleotides with each mutant variant. After incubation, a lower level of remaining oligonucleotides indicated stronger exonuclease activity.

The SRHS mutants SRHS_A, SRHS_B, SRHS_C, and SRHS_D maintained robust exonuclease activity comparable to wild-type (Figure 4A and Appendix A), demonstrating that the unique Lys129-Asp538 interaction did not significantly impact exonuclease function. In contrast, SRHS_E and SRHS_F showed markedly reduced degradation efficiency, suggesting that mutations affecting primer binding (G96R, M97K) and catalytic center residues (D486E, K515Y) impaired exonuclease activity.

For the BBum polymerases, variants BBum_Pro_E, BBum_Pro_F, BBum_Pro_G, BBum_Pro_H, BBum_Pro_I, BBum_Pro_J, BBum_Pro_K, BBum_Pro_L, BBum_Exo_B, and BBum_Exo_C retained wild-type-level exonuclease activity (Figure 4B and Appendix A). However, BBum_Exo_A (D12A) and BBum_Exo_D (D170A) exhibited significantly reduced activity, confirming these residues as critical for exonuclease function—consistent with previous reports of analogous mutations in Phi29 polymerase [29]. Notably, BBum_Exo_B (N63D) and BBum_Exo_C (Y166C) showed no activity reduction, unlike their Phi29 counterparts [39,40]. The BBum_Pro_M variant displayed weak exonuclease activity, likely due to disruption of key functional domains affecting both exonuclease and polymerization capabilities.

### 3.4. Processivity Analysis of SRHS and BBum Mutants

Processivity analysis was conducted to assess the extension capabilities of the SRHS and BBum mutants. For the SRHS variants, SRHS_A and SRHS_B maintained extension ability comparable to wild-type (Figure 5A and Appendix A), indicating that the K129S and D538A mutations do not impair polymerase activity. In contrast, SRHS_C showed reduced DNA product intensity and length, indicating that the Q132K or D538K modifications negatively impacted natural substrate incorporation (Appendix A). SRHS_D demonstrated better polymerase activity than SRHS_C, likely due to the additional S187R and N196K mutations enhancing template binding. In addition, although the length of DNA products for SRHS_E and SRHS_F was significantly reduced, these variants exhibited minimal residual template–primer complex, suggesting enhanced binding affinity to the template–primer complex (Figure 5A and Appendix A). In the case of the mutant sites in SRHS_B, SRHS_E, and SRHS_F, the mutations affecting primer binding affinity (G96R and M97K) and those in the catalytic center (D486E and K515Y) impaired continuous incorporation of dNTPs, leading to the production of shorter DNA products.

Among BBum mutants, only BBum_Exo_B and BBum_Pro_E retained wild-type-level polymerization activity, while other mutants showed reduced incorporation ability (Figure 5B and Appendix A). Although DNA product lengths of BBum_Pro_F, BBum_Pro_G, and BBum_Pro_H, were similar to BBum_wildtype, their product intensities were weaker. This result suggested that the S97R, M98K, and A493E mutations did not improve natural substrate incorporation. Interestingly, BBum_Pro_I, BBum_Pro_J, BBum_Pro_K, and BBum_Pro_L generated shorter DNA products than the BBum_wildtype, but their remaining template–primer complex levels were lower than those of other variants (Figure 5B). This suggests that functional domain replacements targeting residues 510–542 or 516–542 in BBum polymerases may be critical for template–primer complex binding during DNA replication. Additionally, the remaining template–primer complex levels of BBum_Exo_A, BBum_Exo_B, BBum_Exo_C, and BBum_Exo_D were comparable to those of BBum_Pro_I, BBum_Pro_J, BBum_Pro_K, and BBum_Pro_L. However, we proposed that this similarity may result from exonuclease activity. BBum_Pro_M exhibited severely compromised extension capability, likely due to the substitution of polymerase’s functional domain. Furthermore, the incorporation efficiency of BBum_Exo_A, BBum_Exo_C, and BBum_Exo_D was significantly lower than that of BBum_wildtype and other mutant variants, suggesting that the D12A (BBum_Exo_A), Y166C (BBum_Exo_C), and D170A (BBum_Exo_D) mutations inhibited polymerase activity (Figure 5B). This finding was consistent with previous research showing that the mutations D12A, Y165C, or D169A in Phi29 polymerases significantly reduced the polymerization [7,40].

### 3.5. Processivity of Mutant Variants Using Unnatural Substrates

Previous studies of polymerase–nanopore sequencing applications demonstrated the ability of Phi29 DNA polymerases to incorporate polymer-tagged deoxyribonucleotide with Mn^2+^ as the catalytic ion [27,41]. To assess the potential of SRHS and BBum polymerases for single-molecule sequencing application, rolling circle amplification (RCA) was performed using the polymer-tagged nucleotides as substrates. All tested wild-type DNA polymerases, including Phi29, SRHS, and BBum, successfully incorporated unnatural substrates in the presence of Mn^2+^. However, none of these polymerases functioned with Mg^2+^, Ca^2+^, or Sr^2+^ (Figure 6A,B). Notably, Fe^2+^ supported the extension by Phi29, but not by wild-type BBum and SRHS polymerases, suggesting specificity of catalytic ions among the polymerases.

For SRHS mutant variants, all of them successfully amplified DNA using unnatural substrates in the presence of Mn^2+^, as shown in Figure 6C. The DNA product of SRHS_wildtype (SRHS_WT), SRHS_B, and SRHS_E exhibited comparable length, though inferior to Phi29 DNA polymerases. The continuous incorporation efficiency of SRHS_A was lower than SRHS_B and SRHS_WT. This indicated that K129S was a detrimental mutation, whereas D538A was beneficial. Notably, SRHS_C and SRHS_D produced longer DNA products than SRHS_A and SRHS_B, indicating enhanced processivity. This improvement implied that Q132K or D538K mutations facilitated more efficiently continuous extension with unnatural substrates. Furthermore, SRHS_D yielded stronger DNA product intensity and had lower remaining template–primer complex levels than SRHS_C, suggesting that S187R and N196K might enhance amplification efficiency through improving template–primer complex binding. The residual template levels for SRHS_E and SRHS_F were the lowest among all variants, indicating that these mutants exhibited superior template–primer complex utilization efficiency (Figure 6C). Key mutations responsible for enhanced template binding were identified as M97K, D486E, and K515Y. Notably, SRHS_E demonstrated marginally better performance than SRHS_F, suggesting that the S96R mutation does not contribute significantly to functional improvement.

SRHS_D (G96R, M97K, K129S, D486E, K515Y, and D538A) showed the best processivity using Mn^2+^ as a catalytic ion and modified nucleotides as substrates (Figure 6C). To identify critical mutations, we designed additional variants: SRHS_H (K129S, Q132K), SRHS_J (K129S, D538K), and SRHS_K (S187R, N196K). As is shown in Figure 6D, SRHS_K exhibited lower residual template–primer complex levels than SRHS_WT, confirming that S187R and N196K improved template–primer complex binding—consistent with observations in SRHS_C and SRHS_D (Figure 6C,D). A comparison between SRHS_A (K129S), and SRHS_H revealed that Q132K increased DNA product intensity (Figure 6D). SRHS_A and SRHS_J demonstrated that D538K promotes polymerase extension (Figure 6D). Strikingly, SRHS_C and SRHS_H extension patterns suggest that D538K is more effective than Q132K in improving the length of DNA products.

For BBum mutant variants in unnatural substrate conditions, BBum_Pro_E, BBum_Pro_F, BBum_Pro_G, BBum_Pro_H, BBum_Pro_I, BBum_Pro_J, BBum_Pro_K, and BBum_Pro_L demonstrated acceptable polymerization activity, whereas BBum_Pro_M, BBum_Exo_A, BBum_Exo_B, BBum_Exo_C, and BBum_Exo_D exhibited reduced incorporation efficiency (Figure 6E). Comparisons among BBum_Pro_F (A493E), BBum_Pro_G (S97R, M98K), and BBum_Pro_H (S97R, M98K, A493E) suggested that the individual contributions of S97K, M98K, and D483E are minimal, but when combined, these mutations synergistically enhance the polymerase’s ability to incorporate unnatural substrates. BBum_Pro_H, BBum_Pro_I, BBum_Pro_J, BBum_Pro_K, and BBum_Pro_L had lower remaining template–primer complex levels and produced longer and higher-intensity DNA products than other variants. The template–primer binding efficiencies of BBum_Pro_I, BBum_Pro_J, BBum_Pro_K, and BBum_Pro_L were consistent with their performance on natural substrates (Figure 5B and Figure 6E).

### 3.6. Key Mutantions Enabling Mg^2+^-Dependent Unnatural Substrate Incorporation

Mg^2+^ serves as the natural catalytic ion for replicative DNA polymerases. As the predominant physiological cofactor, it ensures high replication fidelity by stabilizing dNTP conformations and promoting correct base pairing. However, its use with polymer-tagged nucleotides has presented significant challenges. Our studies confirmed that wild-type SRHS, BBum, and Phi29 DNA polymerases fail to incorporate unnatural substrates in Mg^2+^-containing buffers (Figure 6A,B). Notably, we engineered several mutant variants capable of overcoming this limitation. SRHS_E, SRHS_F, BBum_Pro_H, BBum_Pro_I, BBum_Pro_J, BBum_Pro_K, and BBum_Pro_L successfully amplified DNA utilizing Mg^2+^ as the catalytic ion, polymer-tagged nucleotides as substrates, and hairpin DNA as templates, though BBum_Pro_J and BBum_Pro_K showed weaker product formation (Figure 7A,B). In contrast, other SRHS variants, BBum variants and Phi29 polymerases were unable to extend under these conditions. Notably, BBum_Pro_L and SRHS_F displayed higher incorporation efficiency, making them prime candidates for polymerase–nanopore sequencing applications.

Fidelity assessment revealed that BBum_Pro_L maintains accuracy comparable to Phi29 polymerase in the presence of Mg^2+^ [42], as determined by our novel isothermal amplification fidelity assay (Appendix A). However, Mn^2+^ as metal cofactors resulted in lower fidelity than Mg^2+^ (Appendix A). Based on commonly shared site analysis of the mutational profiles present in SRHS_E (M97K, K129S, D486E, K515Y, D538A), SRHS_F (G96R, M97K, S187R, N196K, D486E, K515Y), BBum_Pro_H (S97R, M98K, A493E), BBum_Pro_I (S97R, M98K, A493E, (phi29 (507–524) instead of 27(516–542))), BBum_Pro_J (S97R, M98K, A493E, (phi29 (501–524) instead of 27(510–542))), BBum_Pro_K (S97R, M98K, A493E, (phi29 (507–524, K512Y) instead of 27(516–542))), and BBum_Pro_L (S97R, M98K, A493E, (phi29 (501–524, K512Y) instead of 27(510–542)),) (Figure 7A,B and Table 1), we suggested that M97K, G96R, and D486E in SRHS polymerases, as well as M98K, S97R, and A493E in BBum polymerases, are critical residues for Mg^2+^-dependent unnatural substrate incorporation.

To further identify key residues enabling Mg^2+^-dependent incorporation of unnatural substrates, we engineered a series of SRHS_F-derived variants (Table 1). Among these, SRHS_G (D468E, K515Y) demonstrated the ability to incorporate unnatural substrates with Mg^2+^, albeit with weak DNA product yield, suggesting that D486E contributes to catalytic ion selection (Figure 7C). Comparative analysis revealed that M97K significantly enhances extension efficiency, as evidenced by the improved performance of SRHS_I (M97K, D468E, K515Y). Further enhancement was observed in SRHS_L (G96R, M97K, D468E, K515Y), which exhibited superior extension capability compared to SRHS_I (Figure 7C). These results indicated that the G96R mutation facilitates polymerase activity under unnatural substrate/Mg^2+^ conditions, aligning with the results of SRHS_E and SRHS_F (Figure 7A,C). Interestingly, the addition of S187R and N196K (SRHS_M: G96R, M97K, S187R, N196K, D468E, K515Y) did not increase polymerase activity, consistent with the observation that SRHS_K failed to extend products with unnatural substrates in the presence of Mg^2+^ (Figure 7C and Table 1).

When evaluated using primed SSC templates with unnatural substrates and Mg^2+^, distinct patterns emerged: SRHS_G failed to amplify DNA products, while SRHS_I, SRHS_L, and SRHS_M produced robust products (Figure 7D). This confirmed the beneficial roles of G96R and M97K in generating longer products. Notably, SRHS_M exhibited the highest processivity with primed SSC templates (Figure 7D), revealing that S187R and N196K are particularly effective for this template–primer complex architecture.

Collectively, these results established G96R, M97K, and D486E as critical mutations enabling efficient Mg^2+^-dependent incorporation of unnatural substrates, while highlighting the importance of template architecture in polymerase performance.

## 4. Discussion

Microbial DNA polymerases with strand displacement ability have emerged as indispensable tools across biotechnology, molecular biology, and diagnostic applications due to their exceptional fidelity and synthetic efficiency. This study characterized two novel phage-derived enzymes—SRHS from *Bacillus* phage SRT01hs and BBum from *Bacillus* phage BeachBum—demonstrating these polymerases as functional homologs of the well-known Phi29 DNA polymerases. Through AlphaFold structural predictions and comprehensive biochemical assays, both polymerases exhibited the following: (1) robust 3′→5′ exonuclease activity, (2) efficient strand displacement capability, (3) high processivity during DNA synthesis. These Phi29-like properties suggest strong potential for demanding applications including whole-genome amplification and long-read sequencing technologies.

### 4.1. Exonuclease Activity and Structural Implications

Phosphorothioate-modified oligonucleotide degradation assays confirmed the strict 3′→5′ exonuclease specificity of SRHS and BBum polymerases, a hallmark of B family polymerases. This activity aligned with reported characteristics of other strand-displacing B family members like Phi29, T4, Bam35, and RB69 polymerases [3,5,23,43] (Appendix A). Although the exonuclease activity typically targeted double-stranded DNA (dsDNA), our results demonstrated that Phi29, SRHS, and BBum efficiently degraded single-stranded oligonucleotides (25bp). This ssDNA degradation capability—previously employed to confirm exonuclease activity of IME199 DNA polymerases [5]—differed from dedicated exonucleases like Exonuclease I (Exo I). Exo I is a non-processive exonuclease highly specific for ssDNA, whereas SRHS/BBum/Phi29/IME199 are processive enzymes whose exonuclease domain evolved primarily for dsDNA proofreading. We proposed two non-exclusive mechanisms enabling this ssDNA degradation: (1) structural flexibility of the exonuclease site, which accommodated transiently folded ssDNA; (2) kinetic partitioning, where the absence of polymerization shifted the equilibrium toward exonuclease activity [44]. Polymerases naturally balance exonuclease and polymerase activities. However, under our exonuclease assay conditions—which provided Mg^2+^ (essential for catalysis) but lacked dNTP substrates and templates—the absence of polymerization created a strong kinetic bias towards exonuclease activity.

Through targeted mutagenesis, we identified critical residues governing exonuclease function. For example, the impaired exonuclease activity of SRHS_E and SRHS_F implicated residues G96R, M97K, D486E, or K515Y as essential candidate sites for exonuclease performance. Of particular interest, *Bacillus* phage BeachBum exhibits remarkable differences from previously characterized phi29-like phages, such as distinctive genomic features [45]. These differences suggested that BBum may have evolved unique mechanisms for DNA replication, setting it apart from other members of the phi29-like phage family. Mutations D12A and D170A (BBum_Exo_A and BBum_Exo_D) significantly reduced exonuclease activity, mirroring the conserved catalytic roles of analogous residues in Phi29 polymerases (D12 and D169) [23] and in IME199 polymerases (D30 and D251) [5]. However, mutations Y166C (BBum_Exo_C) and N63D (BBum_Exo_D) did not significantly affect exonuclease activity, differing from the functional roles of Y165C and N62D in Phi29 polymerase [23], suggesting differences in active sites between BBum and Phi29 polymerases. These findings highlighted both conserved and divergent functional elements within the exonuclease domains of these polymerases.

### 4.2. Processivity and Strand Displacement Activity

We evaluated polymerase processivity and strand displacement capability using rolling circle amplification (RCA) assays with primed single-stranded circular (SSC) templates. These assays distinguished highly processive polymerases, which generated long, high-molecular-weight replication products (visible as smears or high-mass bands on agarose gels), from less processive enzymes that produce shorter, discrete fragments due to frequent template dissociation. The presence of long replication products, visualized by agarose gel, indicated sustained processivity and efficient strand displacement. In contrast, truncated products or low yields suggested limited processivity or dissociation. In addition, the remaining template–primer complex on the agarose gel reflected the utilization efficiency of polymerases for the template–primer complex. The fewer remaining templates indicated that more templates were used for extension.

Processivity assays (Figure 2) demonstrated that both SRHS and BBum polymerases maintained high strand displacement activity across a broad temperature range (10–45 °C) and with various catalytic ions (Mg^2+^, Mn^2+^ or Fe^2+^), a hallmark of Phi29-like polymerases. These findings aligned with the established literature showing Mg^2+^ and Mn^2+^ as effective cofactors for strand-displacing B family DNA polymerases [46,47]. Interestingly, our results showed that Ca^2+^ failed to support polymerase activity of strand-displacing B family DNA polymerases Phi29, SRHS, and BBum. But for strand-displacing A family DNA polymerase Bst, Ca^2+^ can still catalyze its extension [16]. This reveals the difference in cofactor requirements between distinct DNA polymerase families.

Further mutational analysis of SRHS and BBum polymerases provided further insights into their processivity in dNTP conditions. In SRHS polymerases, mutations in the primer-binding region (G96R, M97K) and catalytic center (D486E) reduced processivity but enhanced template–primer complex utilization efficiency. Parallel observations in BBum variants (S97R, M98K, and A493E) also confirmed that these residues are crucial for incorporation ability. These results are consistent with previous studies identifying analogous residues such as Arg96 and Asp458 as critical for template–primer binding and polymerase activity in Phi29 polymerases [48,49]. Interestingly, functional domain replacements with Phi29-derived sequences (residues 510–542 and 516–542) affected processivity of BBum polymerases, suggesting that modifications in these regions impacted template–primer complex binding. Fusing functional domains such as sulfolobus 7 kDa protein sso7d or the helix–hairpin–helix [(HhH)_2_] domain to DNA polymerases represents an established strategy for enhancing processivity [50,51]. Although these fusion domains (e.g., Sso7d) improved processivity via augmented DNA-binding affinity, their structural compatibility with SRHS/BBum polymerases remains unvalidated. Critically, in polymerase–nanopore sequencing systems, the elongated linker between the polymerase and nanopores—resulting from domain fusion—may perturb enzyme–pore coupling dynamics during long-read sequencing.

### 4.3. Processivity with Unnatural Substrates

In single-molecule sequencing systems, DNA polymerases are required to incorporate oligonucleotide-tagged dNTPs. During DNA synthesis, each incorporation released a tag that passes through the nanopore and generated a base-specific electrical signal. To distinguish all four bases, four different modified nucleotides were used [22]. Phi29 polymerase has been shown to incorporate polymer-tagged nucleotides using Mn^2+^ as a catalytic ion, enabling its use in polymerase–nanopore sequencing [22]. Our results demonstrate that specific mutations in SRHS (Q132K, S187R, N196K, and D538K) and BBum (S97R, M98K, A493E) polymerases achieved comparable functionality. This underscores their potential for single-molecule sequencing applications.

Mn^2+^ as catalytic ions enable the incorporation of unnatural substrates that are not efficiently incorporated with Mg^2+^ [19], likely due to its larger ionic radius (0.83 Å compared to Mg^2+^’s 0.73 Å) and more flexible coordination geometry [17]. This flexibility allowed polymerases to accommodate structural deviations in the substrate or active site, enabling the incorporation of unnatural nucleotides. However, Mn^2+^ is commonly used in error-prone PCR to increase the error rate of DNA polymerases, making the technique highly effective for creating mutant libraries [52]. This indicated that Mn^2+^ was unsuitable for applications requiring high fidelity, such as nanopore sequencing. Remarkably, our study revealed that specific SRHS and BBum mutants (SRHS_E, SRHS_F, BBum_Pro_H-L) can incorporate unnatural substrates using Mg^2+^ as the catalytic ion, a capability not observed in wild-type polymerases or Phi29. These results suggested that mutations such as G96R, M97K, and D486E (SRHS) and S97R, M98K, and A493E (BeachBum) were critical for expanding polymerase substrate compatibility in the presence of Mg^2+^. These mutations may facilitate improved polymerase–Mg^2+^ coordination and stabilize primer–template binding, thereby enabling efficient polymerization with challenging substrates. Notably, these adaptive mutations appeared specific to unnatural substrates, as they do not enhance incorporation efficiency with natural nucleotides.

Overall, our findings significantly expand the understanding of SRHS and BBum polymerases and their potential applications in single-molecule sequencing systems. Their Mg^2+^-dependent activity with unnatural substrates distinguishes them from other strand-displacing B family polymerases such as Phi29 and suggests promising avenues for enzyme optimization. Future studies should explore the engineering of these enzymes to optimize their substrate specificity, fidelity, and processivity, which could enhance their utility in diagnostic and single-molecule sequencing systems.

## 5. Conclusions

This study successfully characterized two novel *Bacillus* phage DNA polymerases, SRHS and BBum, demonstrating their structural and functional homology with the well-established Phi29 polymerases, despite sharing limited sequence identity (~55%). Through biochemical analysis and structural characterization using AlphaFold3-predicted models, both enzymes exhibit robust phi29-like properties, including high processivity, efficient strand displacement, and strong 3′→5′ exonuclease activity with minimal 5′→3′ activity, across physiologically relevant temperatures (10–45 °C) and ionic conditions (Mg^2+^, Mn^2+^, Fe^2+^). AlphaFold-guided mutagenesis identified critical residues and domains regulating these functions. Our key findings include the following: (1) Asp12 and Asp170 were essential for BBum exonuclease activity, mirroring Phi29; (2) mutations in SRHS (G96R, M97K, D486E) and BBum (S97R, M98K, A493E) enabled unprecedented incorporation of polymer-tagged unnatural nucleotides using Mg^2+^, a capability lacking in wild-type strand-displacing B family enzymes; (3) specific mutations (e.g., SRHS S187R/N196K, BBum domain swaps in Pro_I/L variants) enhanced template–primer binding and processivity, especially with unnatural substrates.

These engineered polymerases, particularly SRHS_F and BBum_Pro_L, represent superior alternatives to Phi29 for applications requiring Mg^2+^-dependent synthesis of modified nucleotides. This capability is essential for polymerase–nanopore sequencing, which utilizes oligonucleotide-modified nucleotides as substrates to generate accurate sequencing signals. Our study elucidates structure–function relationships in phage strand-displacing B family polymerases and provides a platform for further engineering tailored enzymatic properties. Future efforts will focus on improving the length of the synthesized DNA products from unnatural substrates under Mg^2+^ conditions to advance the read length of nanopore sequencing systems.

## Figures and Tables

**Figure 1 biomolecules-15-01126-f001:**
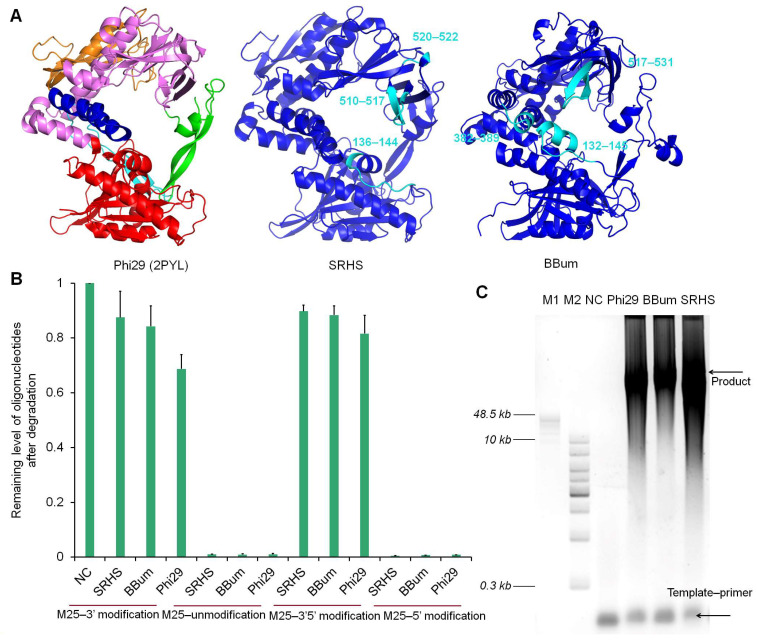
The exonuclease activity and processivity analysis of the SRHS and BBum polymerases. (**A**) The AlphaFold3-predicted structures of the SRHS (pTM = 0.95) and BBum (pTM = 0.95) polymerases exhibited overall topological similarity to the experimentally determined structure of Phi29 (PDB: [2PYL]). A pTM > 0.95 indicated an expected structural deviation of less than 5% in conserved domains compared to experimental data. The Phi29 structure was color-coded as follows: the exonuclease domain (residues 1–190) in red; palm domain (residues 191–261 and 428–531) in pink; TPR1 domain (residues 262–359) in orange; fingers domain (residues 360–395) in blue; TPR2 domain (residues 396–427) in cyan; and thumb domain (residues 532–575) in green. In the predicted structures, regions with high pLDDT scores (>90) are colored blue, while regions with moderate pLDDT scores (70–90) are colored cyan. (**B**) The exonuclease activity assays of various polymerases under four oligonucleotide conditions: M25 (unmodified), M25-3′ (with three consecutive phosphorothioate (PS) modification at the 3′ terminal), M25-5′ (with three consecutive PS modifications at the 5′ terminal), and M25-3′5′ (with PS modifications at both terminals). The reactions were conducted using 1 µM oligonucleotides and 200 nM of polymerase in 1× phi29 reaction buffer. The mixtures were incubated at 30 °C for 30 min, followed by analysis using 10% TBE-Urea gel and quantification with Image J. Levels were normalized to the negative control (NC) which just added oligonucleotides. Three constitutive PSs as a protecting group can protect oligonucleotides from degrading. (**C**) Rolling circle amplification (RCA) assays revealed the high processivity of SRHS, BBum, and Phi29. In total, a 100 nM primed template was incubated with 100 nM polymerases and 100 μM dNTPs in Mg^2+^-containing buffer at 30 °C for 1 h. The result was then analyzed through 0.6% agarose gel electrophoresis. NC refers to negative control which just possessed templates. M1 refers to the 1 KB Plus DNA Ladder; M2 refers to the GeneRuler High Range DNA Ladder.

**Figure 2 biomolecules-15-01126-f002:**
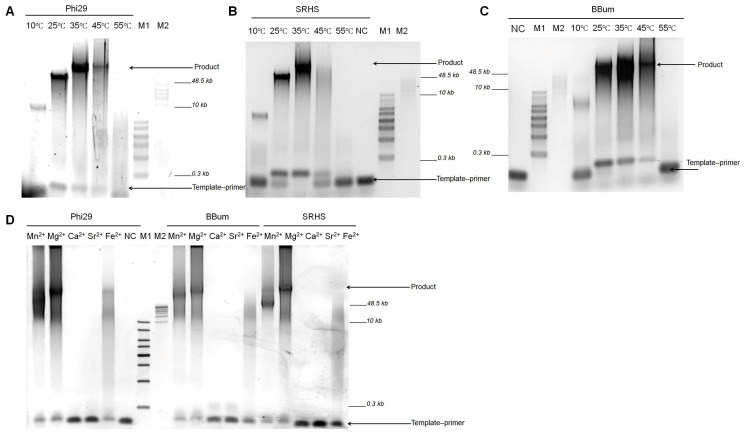
The biochemical characteristics of the Phi29, SRHS, and BBum polymerases. (**A**–**C**) The effect of temperature on the polymerase activity of the Phi29, SRHS, and BBum polymerases. A 100 nM primed template was incubated with 100 nM polymerases and 100 μM dNTPs at different temperatures for 1 h in Mg^2+^-containing reaction buffer. The results were analyzed with 0.6% agarose gel electrophoresis. (**D**) The effect of different metal ions on the polymerase activity of the Phi29, SRHS, and BBum polymerases. Reactions were performed with a 100 nM primed template, 100 nM polymerase, and 100 μM dNTPs at 30 °C for 1 h in reaction buffers supplemented with different divalent metal ions (1 mM final concentration), and products were analyzed by 0.6% agarose gel electrophoresis. NC refers to negative control which just possessed templates. M1 refers to the 1 KB Plus DNA Ladder; M2 refers to the GeneRuler High Range DNA Ladder.

**Figure 3 biomolecules-15-01126-f003:**
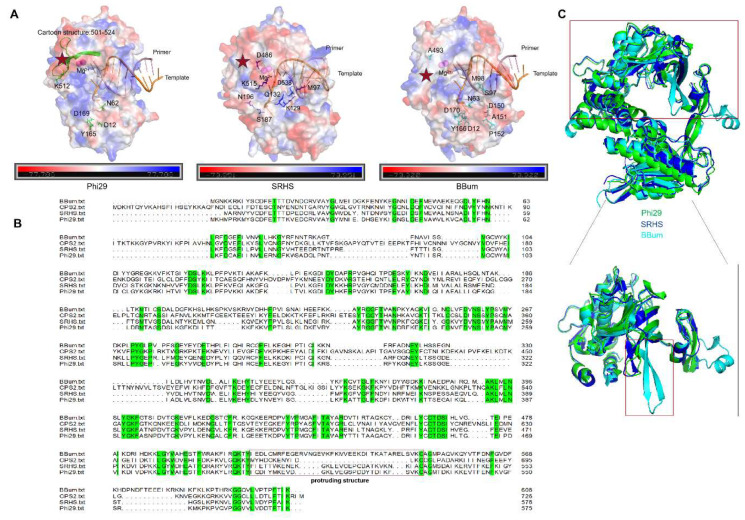
The mutant sites and structure domain analysis of the Phi29, SRHS, and BBum polymerases. (**A**) The mutant sites in the SRHS and BBum mutant variants. The red stars refer to the protruding structures near the substrate entrance of DNA polymerases. (**B**) The sequence alignments of the CPS2, Phi29, SRHS, and BBum polymerases. Green indicates identical amino acids; The red box refers to different lengths of the protruding structures near the substrate entrance in the CPS2, Phi29, SRHS, and BBum polymerases. (**C**) The structure alignments of the Phi29, SRHS, and BBum polymerases showed a distinctive protruding structure near the substrate entrance in the BBum polymerase compared with the Phi29 and SRHS polymerases.

**Figure 4 biomolecules-15-01126-f004:**
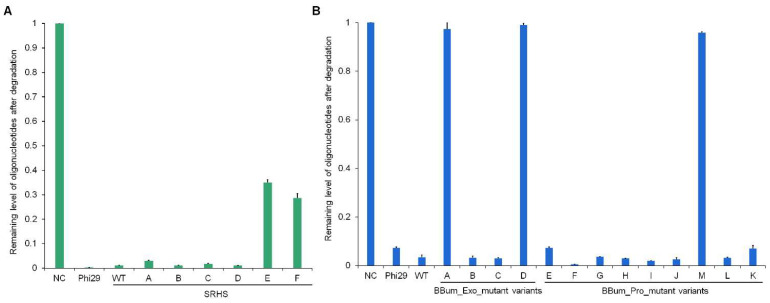
The exonuclease activity analysis of SRHS and BBum mutant variants. (**A**,**B**) Exonuclease activity analysis of polymerase mutant variants. We mixed 1 µM M25-5′ oligonucleotides and 200 nM polymerase mutant variants in a 1 × phi29 buffer at 30 °C for 30 min. The results were analyzed by 10% TBE-Urea gel electrophoresis and quantified by ImageJ (version 2.35; National Institutes of Health, Bethesda, MD, USA). Levels were normalized to the negative control (NC) which just added oligonucleotides.

**Figure 5 biomolecules-15-01126-f005:**
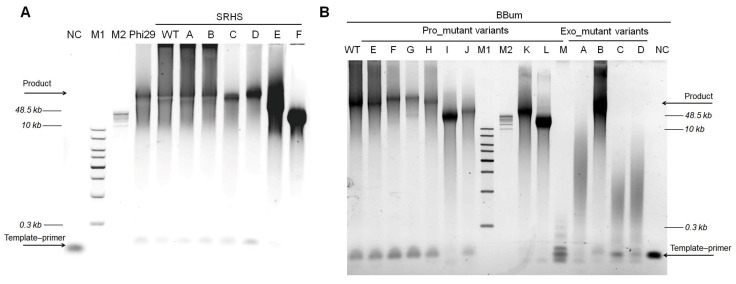
The processivity analysis of SRHS and BBum mutant variants. (**A**,**B**) We incubated a 100 nM primed template with 100 nM polymerases and 100 μM dNTPs for 1 h. The results were then analyzed on 0.6% agarose gel electrophoresis. M1 refers to the 1 KB Plus DNA Ladder; M2 refers to the GeneRuler High Range DNA Ladder. NC refers to the negative control which just added the primed template.

**Figure 6 biomolecules-15-01126-f006:**
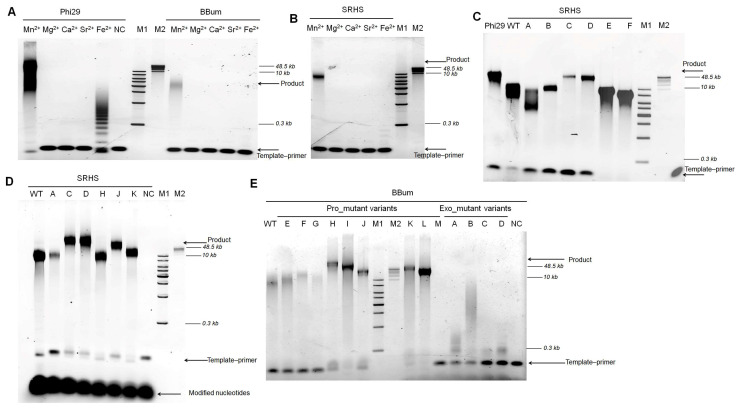
The processivity of SRHS and BBum polymerases with unnatural substrates. (**A**,**B**) The effect of metal cofactors on DNA extension. We incubated a 100 nM primed template with 100 nM polymerases and 100 μM unnatural substrates at 30 °C for 1 h in reaction buffers containing different metal ions (1 mM final concentration). The results were then analyzed through 0.6% agarose gel electrophoresis. (**C**–**E**) The extension efficiency of mutant variants in unnatural substrate and Mn^2+^ conditions. We incubated a 100 nM primed template with 100 nM polymerases and 100 μM unnatural substrates at 30 °C for 1h in reaction buffers. The results were then analyzed through 0.6% agarose gel electrophoresis. M1 referred to the 1 KB Plus DNA Ladder; M2 referred to the GeneRuler High Range DNA Ladder. NC refers to the negative control which just added the primed template and unnatural substrates.

**Figure 7 biomolecules-15-01126-f007:**
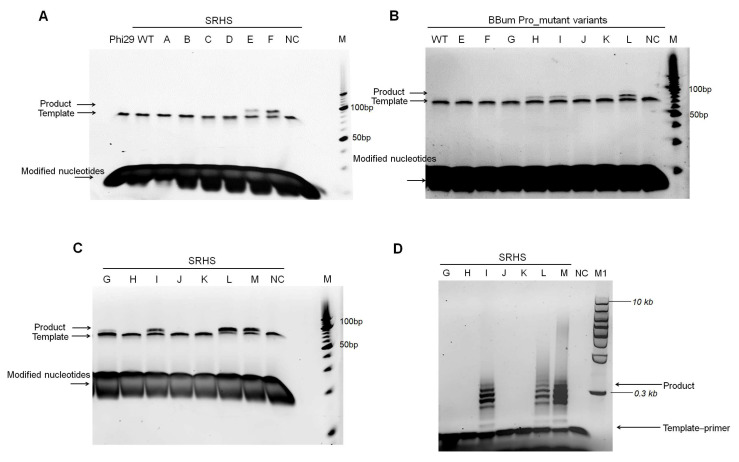
The processivity of SRHS and BBum polymerases with unnatural substrates and Mg^2+^. (**A**–**C**) Extension ability analysis of mutant variants with a hairpin template. The 20 nM hairpin template was incubated with 10 nM polymerases and 100 μM unnatural substrates at 30 °C for 30 min in Mg^2+^ -containing buffer. The results were then analyzed on 10% TBE-Urea gel. M refers to the O’RangeRuler 10 bp DNA Ladder. NC refers to the negative control with templates and unnatural substrates. (**D**) Extension ability analysis of mutant variants with primed SSC templates. We incubated a 100 nM primed template with 100 nM polymerases, 100 μM unnatural substrates, and 1 mM Mg^2+^ at 30 °C for 1h. The results were then analyzed on 0.6% agarose gel electrophoresis. M1 refers to the 1 KB Plus DNA Ladder. NC refers to the negative control which just added the primed template and unnatural substrates.

**Table 1 biomolecules-15-01126-t001:** Mutant sites of SRHS and BBum polymerase mutant variants.

Mutant Name	Mutant Sites
SRHS_A	K129S
SRHS_B	K129S, D538A
SRHS_C	K129S, Q132K, D538K
SRHS_D	K129S, Q132K, S187R, N196K, D538K
SRHS_E	M97K, K129S, D486E, K515Y, D538A
SRHS_F	G96R, M97K, K129S, D486E, K515Y, D538A
SRHS_G	D486E, K515Y
SRHS_H	K129S, Q132K
SRHS_I	M97K, D486E, K515Y
SRHS_J	K129S, D538K
SRHS_K	S187R, N196K
SRHS_L	G96R, M97K, D486E, K515Y,
SRHS_M	G96R, M97K, S187R, N196K, D486E, K515Y,
BBum_Exo_A	D12A
BBum_Exo_B	N63D
BBum_Exo_C	Y166C
BBum_Exo_D	D170A
BBum_Pro_E	D150H, A151K, P152E
BBum_Pro_F	A493E
BBum_Pro_G	S97R, M98K,
BBum_Pro_H	S97R, M98K, A493E
BBum_Pro_I	S97R, M98K, A493E, (phi29 (507–524) instead of 27(516–542))
BBum_Pro_J	S97R, M98K, A493E, (phi29 (501–524) instead of 27(510–542))
BBum_Pro_K	S97R, M98K, A493E, (phi29 (507–524, K512Y) instead of 27(516–542))
BBum_Pro_L	S97R, M98K, A493E, (phi29 (501–524, K512Y) instead of 27(510–542))
BBum_Pro_M	S97R, M98K, A493E, (CPS2 (662–674) instead of 27(510–542))

## Data Availability

Representative AlphaFold3 models are available at https://www.modelarchive.org/. SRHS: https://www.modelarchive.org/doi/10.5452/ma-4qoln (accessed on 30 July 2025), access code: jJL3ZiUP0O. BBum: https://www.modelarchive.org/doi/10.5452/ma-0odsb (accessed on 28 July 2025), access code: ZUkVpQUXRb.

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
