# Peer review of "Characterization and Engineering of Two Novel Strand-Displacing B Family DNA Polymerases from Bacillus Phage SRT01hs and BeachBum"

_biomolecules, 2025, doi:10.3390/biom15081126_

Round 1

Reviewer 1 Report

Comments and Suggestions for Authors

The authors report the study of two novel family B DNA polymerases from Bacillus phages. It was shown that both enzymes exhibit robust strand displacement, 3′→5′ exonuclease activity across a temperature range from 15°C to 45°C and under various catalytic ion conditions. Structural prediction and comparative modeling allowed to create mutant variants of enzymes with modified properties. Authors concluded that both enzymes are promising candidates for precision DNA synthesis using unnatural substrates, with significant implications for genome engineering and next-generation sequencing technologies.

There are some major comments to the obtained results.

  • Authors concluded that SRHS and BBum could be used as next-generation enzymes for precision biotechnological applications. However, many of DNA polymerases of family B are quite well studied in the terms of mechanism of action, catalytic efficiency, processivity and substate selectivity, and some of these enzymes (including but not limited to the enzyme Phi29) are already widely used in multiple applications. If the authors focus on specific molecular biological applications of the enzymes they create (for example, polymerases for isothermal amplification, rolling circle amplification, DNA synthesis using unnatural dNTPs, etc), it will not raise so many questions and expectations.
  • A comparative analysis of properties of novel enzymes with widely used enzymes (including literature data) should be discussed. Such comparison could be made by dividing results in different properties, which are important for exact application (probably in the table format). For example, activity in a temperature range from 15°C to 45°C is relevant for isothermal amplification but is not as wide as known for some other enzymes of this family, which are used in PCR amplification.
  • The data on processivity are not clear and do not allow to make conclusion because of “more” or “less” intensity of bands in the Fig. 5 can be used only for visual analysis, but presented results do not allow to obtain a quantitative characteristic that can be compared for different enzymes and/or for different experimental conditions. As well as, if processivity is important enzyme parameter, discussion of creation of phusion enzyme with Sso7d or other domains also could be mentioned.
  • Authors state that their enzymes can be used for precision biotechnological applications, but in the manuscript and SI no data (number of blue and white clones or other quantitative characteristics) supporting precision of DNA synthesis or fidelity of these enzymes.
  • Five different cofactor ions were used in the study. However, the basis for the choice of these metals is not described. What the authors expected when using Ca²⁺ and Sr²⁺?
  • The structure and significance of using unnatural substrates for these enzymes is not clear, why their use is required, for what purpose, whether something else can be used for this purpose, etc. This information should be briefly precented in the manuscript to make understanding of results clearer.
  • The main disadvantage of this work for readers is that it is very difficult to see the difference between new enzymes and already known ones. And this disadvantage should not be hidden by repeating similar phrases about how enzymes can be used in a variety of biotechnological applications.

Author Response

Response to Reviewer

We sincerely thank you for your valuable and insightful comments that have significantly helped improve the quality of our manuscript. We have carefully addressed all the points, providing point-by-point responses in the revised file. The sections marked in green in both the revised manuscript and this response letter indicate the content that has been modified or newly added in response to the comments.

The authors report the study of two novel family B DNA polymerases from Bacillus phages. It was shown that both enzymes exhibit robust strand displacement, 3′→5′ exonuclease activity across a temperature range from 15°C to 45°C and under various catalytic ion conditions. Structural prediction and comparative modeling allowed to create mutant variants of enzymes with modified properties. Authors concluded that both enzymes are promising candidates for precision DNA synthesis using unnatural substrates, with significant implications for genome engineering and next-generation sequencing technologies.

There are some major comments to the obtained results.

  • Authors concluded that SRHS and BBum could be used as next-generation enzymes for precision biotechnological applications. However, many of DNA polymerases of family B are quite well studied in the terms of mechanism of action, catalytic efficiency, processivity and substate selectivity, and some of these enzymes (including but not limited to the enzyme Phi29) are already widely used in multiple applications. If the authors focus on specific molecular biological applications of the enzymes they create (for example, polymerases for isothermal amplification, rolling circle amplification, DNA synthesis using unnatural dNTPs, etc), it will not raise so many questions and expectations.

Response:

We appreciate this important point and agree that the scope of our claims should be better aligned with the specific features and potential applications of our enzymes. In the revised manuscript, we have clarified our conclusions and now explicitly emphasize the potential of SRHS and BBum polymerases for polymerase-nanopore sequencing applications where their strand-displacement activity, cofactor versatility and unnatural nucleotide incorporation are particularly advantageous. We also emphasize that our study focuses on the Family B DNA polymerases with strand displacement. Generalized language regarding “precision biotechnology” has been replaced with more specific examples.

“In polymerase-coupled nanopore long-read sequencing systems, DNA polymerases must possess strand displacement activity and be capable of incorporating oligonucleotide-tagged dNTPs to generate continuous sequencing signals. Strand-displacing B family DNA polymerases from bacteriophages (e.g., phi29) are widely employed in biotechnology due to their high processivity and fidelity.”

“Consequently, the discovery and engineering of novel B family DNA polymerases with robust strand displacement activity remain an important research priority.”

“These results significantly advance our understanding of polymerase engineering and highlight the potential of SRHS and BBum as next-generation enzymes for applications like nanopore sequencing.”

“Microbial DNA polymerases with strand displacement ability have emerged as indispensable tools across biotechnology, molecular biology, and diagnostic applications due to their exceptional fidelity and synthetic efficiency.”

“In single-molecule sequencing systems, DNA polymerases were required to in-corporate oligonucleotide-tagged dNTPs. During DNA synthesis, each incorporation released a tag that passes through the nanopore and generated a base-specific electrical signal. To distinguish all four bases, four different modified nucleotides were used [22].”

“Overall, our findings significantly expand the understanding of SRHS and BBum polymerases and their potential applications in single-molecule sequencing system. Their Mg²⁺-dependent activity with unnatural substrates distinguishes them from other strand displacing B family polymerases such as Phi29 and suggests promising avenues for enzyme optimization.”

“These engineered polymerases, particularly SRHS_F and BBum_Pro_L, represent superior alternatives to Phi29 for applications requiring Mg²⁺-dependent synthesis of modified nucleotides. This capability is essential for polymerase-nanopore sequencing, which utilizes oligonucleotide-modified nucleotides as substrates to generate accurate sequencing signals. Our study elucidates structure-function relationships in phage strand displacing B-family polymerases and provides a platform for further engineering tailored enzymatic properties.”

  • A comparative analysis of properties of novel enzymes with widely used enzymes (including literature data) should be discussed. Such comparison could be made by dividing results in different properties, which are important for exact application (probably in the table format). For example, activity in a temperature range from 10°C to 45°C is relevant for isothermal amplification but is not as wide as known for some other enzymes of this family, which are used in PCR amplification.

Response:

Thank you for this suggestion. In accordance with the reviewer’s suggestion, we have now added a new Supplementary Table 1 that compares the key properties of SRHS and BBum polymerases with reported strand-displacing family B enzymes, including Phi29, IME99, and Bam35, based on both our data and relevant literature. The comparison includes temperature activity range, strand displacement capability, and cofactor tolerance.

However, in our effort to identify and examine other strand-displacement-active B-family DNA polymerases through literature retrieval, we found that systematic studies specifically targeting this group remain extremely scarce. Among these, Phi29 DNA polymerase is currently the most thoroughly characterized member.

Name

3'-5'Exonuclease activity

Strand Displacement

Processivity

Temperature

Catalytic bivalent metal cations (dNTPs as substrates)

Phi29

Yes

Yes

10℃-45℃

Mg2+, Mn2+, Fe2+

BBum

Yes

Yes

similar to Phi29

10℃-45℃

Mg2+, Mn2+, Fe2+

SRHS

Yes

Yes

similar to Phi29

10℃-45℃

Mg2+, Mn2+, Fe2+

IME199[5]

Yes

Yes

similar to Phi29

15℃-35℃

Mg2+, Mn2+, Ca2+

Bam35[3]

Yes

Yes

Unknown (unreported)

37℃ (unreported range)

Mg2+ (unreported other tests)

  • The data on processivity are not clear and do not allow to make conclusion because of “more” or “less” intensity of bands in the Fig. 5 can be used only for visual analysis, but presented results do not allow to obtain a quantitative characteristic that can be compared for different enzymes and/or for different experimental conditions. As well as, if processivity is important enzyme parameter, discussion of creation of phusion enzyme with Sso7d or other domains also could be mentioned.

Response:

Thank you for your suggestion. We have now included a semi-quantitative densitometry analysis of the gel images using ImageJ (presented in revised Supplementary Figure 8), estimating the average extension intensity. However, for processivity, we still want to show the data with two areas containing the length and the intensity, we still keep the pictures of gels in Figure 5.

Additionally, we discuss the potential for fusing DNA-binding domains such as Sso7d, as successfully demonstrated in the design of Phusion polymerase, as a possible next step to further enhance the processivity of SRHS and BBum in the Discussion section.

“Fusing functional domains such as sulfolobus 7-kDa protein sso7d or helix-hairpin-helix [(HhH)2] domain to DNA polymerases represented an established strategy for enhancing processivity [50, 51]. Although these fusion domains (e.g., Sso7d) improved processivity via augmented DNA-binding affinity, their structural compatibility with SRHS/BBum polymerases remains unvalidated. Critically, in polymerase-nanopore sequencing systems, the elongated linker between polymerase and nanopore—resulting from domain fusion—may perturb enzyme-pore coupling dynamics during long-read sequencing.”

  • Authors state that their enzymes can be used for precision biotechnological applications, but in the manuscript and SI no data (number of blue and white clones or other quantitative characteristics) supporting precision of DNA synthesis or fidelity of these enzymes.

Response:

Thank you for this critical comment. In response to the reviewer's request, we have added the corresponding experimental measurements for the BBum_Pro_L and Phi29 DNA polymerases described in the manuscript to the Supplementary File 1.

“Fidelity assessment revealed BBum_Pro_L maintains accuracy comparable to Phi29 polymerase in the presence of Mg2+ [42], as determined by our novel isothermal amplification fidelity assay (Supplementary File 1). However, Mn2+ as metal cofactors resulted in lower fidelity than Mg2+(Supplementary File 1).”

“Table 7. The results of blue and white colonies for BBum_Pro_L and Phi29 polymerases

Sample Name/Catalytic ions

White colonies + Blue colonies

Blue colonies

White colonies

BBum_Pro_L_replicate 1  (Mg2+)

1927

1926

1

BBum_Pro_L_replicate 2  (Mg2+)

2437

2437

0

BBum_Pro_L_replicate 3  (Mg2+)

1976

1974

2

BBum_Pro_L_replicate 4  (Mg2+)

2558

2557

1

BBum_Pro_L_replicate 5  (Mg2+)

2015

2015

0

BBum_Pro_L_replicate 6  (Mg2+)

1835

1835

0

BBum_Pro_L_replicate 7  (Mg2+)

2032

2031

1

BBum_Pro_L_replicate 8  (Mg2+)

2163

2162

1

Phi29_replicate 1  (Mg2+)

2471

2470

1

Phi29_replicate 2  (Mg2+)

2600

2599

1

Phi29_replicate 3  (Mg2+)

2459

2459

0

Phi29_replicate 4  (Mg2+)

2846

2846

0

Phi29_replicate 5  (Mg2+)

2070

2070

0

Phi29_replicate 6  (Mg2+)

2258

2258

0

Phi29_replicate 7  (Mg2+)

2055

2054

1

Phi29_replicate 8  (Mg2+)

2491

2490

1

Phi29_replicate 1  (Mn2+)

2338

2334

4

Phi29_replicate 2  (Mn2+)

2209

2204

5

Phi29_replicate 3  (Mn2+)

1920

1918

2

Phi29_replicate 4  (Mn2+)

2557

2550

7

Phi29_replicate 5  (Mn2+)

2401

2392

9

Phi29_replicate 6  (Mn2+)

1106

1102

4

Phi29_replicate 7  (Mn2+)

1021

1015

6

Phi29_replicate 8  (Mn2+)

1051

1048

3

Based on the above results (Table 7), we obtained the mean value across all replicates and calculated fidelity (Table 8).

Table 8. The Fidelity of BBum_Pro_L and Phi29 polymerases

Sample Name

Mean (White colonies + Blue colonies)

Mean (Blue colonies)

Mean (White colonies)

Fidelity

BBum_Pro_L (Mg2+)

2117.875

2117.125

0.75

5.32524E-07

Phi29 (Mg2+)

2406.25

2405.75

0.5

3.12469E-07

Phi29 (Mn2+)

1825.375

1820.375

5

4.11904E-06

  • Five different cofactor ions were used in the study. However, the basis for the choice of these metals is not described. What the authors expected when using Ca²⁺ and Sr²⁺?

Response:
Thank you for this critical comment.

We added some explanation of the basis for the choice of these metals in the Results section.

“DNA polymerases required divalent metal ions as catalytic cofactors to facilitate ex-tension. We assessed the metal cofactor selection by RCA with various metal ions (Mg²⁺, Mn²⁺, Fe²⁺, Ca²⁺ and Sr²⁺).These metal ions possess distinct ionic radii: Mg²⁺ (0.72 Å), Mn²⁺ (0.83 Å), Fe²⁺ (0.78 Å), Ca²⁺ (1.00 Å), and Sr²⁺ (1.18 Å) [36]. Mg²⁺ and Mn²⁺ serve as primary catalytic cofactors for DNA polymerases due to their optimal size for precise transition-state stabilization in phosphoryl transfer reactions. Ca²⁺ and Sr²⁺—with significantly larger radii—act as negative controls, as their geometric mismatch disrupts active-site coordination and prevents catalysis. Both ions were often used in nanopore sequencing systems to produce stagnation signals [22, 37]. Although Fe²⁺ has an ionic radius comparable to that of Mg²⁺ and Mn²⁺, its potential role as a catalytic cofactor for DNA polymerase remains unclear.”

  • The structure and significance of using unnatural substrates for these enzymes is not clear, why their use is required, for what purpose, whether something else can be used for this purpose, etc. This information should be briefly presented in the manuscript to make understanding of results clearer.

Response:
We thank the reviewer for this valuable comment. We have now added a brief explanation in the revised manuscript to clarify the purpose of using unnatural nucleotide substrates in the abstract and discussion sections.

“Previous studies of polymerase-nanopore sequencing application demonstrated the ability of Phi29 DNA polymerases to incorporate polymer-tagged deoxyribonucleotide with Mn2+ as the catalytic ion [27, 41]”

“As the predominant physiological cofactor, it ensures high replication fidelity by stabilizing dNTP conformations and promoting correct base pairing.”

“Mg²⁺ serves as the natural catalytic ion for replicative DNA polymerases. As the predominant physiological cofactor, it ensures high replication fidelity by stabilizing dNTP conformations and promoting correct base pairing. However, its use with polymer-tagged nucleotides has presented significant challenges. Our studies confirmed that wild-type SRHS, BBum, and Phi29 DNA polymerases fail to incorporate unnatural substrates in Mg²⁺-containing buffers.”

“In single-molecule sequencing systems, DNA polymerases were required to incorporate oligonucleotide-tagged dNTPs. During DNA synthesis, each incorporation released a tag that passes through the nanopore and generated a base-specific electrical signal. To distinguish all four bases, four different modified nucleotides were used [22].”

  • The main disadvantage of this work for readers is that it is very difficult to see the difference between new enzymes and already known ones. And this disadvantage should not be hidden by repeating similar phrases about how enzymes can be used in a variety of biotechnological applications.

Response:

We appreciate this thoughtful critique. In this article, we provide two new polymerases with strand displacement from B-family DNA polymerases. The biggest advantage of these polymerase mutant variants is the ability of incorporating polymer-tagged unnatural nucleotides in the presence of Mg²⁺—a capability not observed in Phi29 or other wild-type B family polymerases with strand displacement.

In the revised manuscript, we emphasize the requirement of the ability to incorporate polymer-tagged unnatural nucleotides using Mg2+ in nanopore sequencing systems for DNA polymerases and narrowed the range of B-family DNA polymerases to these strand displacing B family polymerases.

For nanopore sequencing application, the disadvantage of these polymerases is the length of the synthesized products generated with unnatural substrates under Mg²⁺ conditions which decided the read length of nanopore sequencing system. We added the point to the Conclusion section.

“These engineered polymerases, particularly SRHS_F and BBum_Pro_L, represent superior alternatives to Phi29 for applications requiring Mg²⁺-dependent synthesis of modified nucleotides. This capability is essential for polymerase-nanopore sequencing, which utilizes oligonucleotide-modified nucleotides as substrates to generate accurate sequencing signals. Our study elucidates structure-function relationships in phage strand displacing B-family polymerases and provides a platform for further engineering tailored enzymatic properties. Future efforts will focus on improving the length of the synthesized DNA products from unnatural substrates under Mg²⁺ conditions to advance the read length of nanopore sequencing system.”

Reviewer 2 Report

Comments and Suggestions for Authors

The authors have developed a new DNA polymerases with strand-displacement activity for isothermal amplification of nucleic acids. Various amplification techniques are widely used in many applications. It is important to develop enzymes that provide specific and highly sensitive detection of DNA or RNA. Thus, the problem that the authors dealt with is very relevant.

The manuscript includes data on the engineering and some properties of two novel DNA polymerases from Bacillus phage SRT01hs and Bacillus phage 3 BeachBum. The work seems very interesting and useful. However, to improve the quality of the manuscript, the authors should provide additional information. For this, the following issues need to be solved:

  1. In Introduction section, when describing isothermal amplification techniques, strand-displacement DNA polymerases and metal cofactors, it will be useful to cite some recent studies, e.g., the follows:

1) Sakhabutdinova AR, Garafutdinov RR. Mechanism of DNA multimerization caused by strand-displacement DNA polymerases. Anal Biochem. 2025 Aug;703:115876. doi: 10.1016/j.ab.2025.115876. 
2) Garafutdinov RR, Gilvanov AR, Kupova OY, Sakhabutdinova AR. Effect of metal ions on isothermal amplification with Bst exo- DNA polymerase. Int J Biol Macromol. 2020 Oct 15;161:1447-1455. doi: 10.1016/j.ijbiomac.2020.08.028. 

3) Vashishtha AK, Konigsberg WH. The effect of different divalent cations on the kinetics and fidelity of Bacillus stearothermophilus DNA polymerase. AIMS Biophys. 2018;5(2):125-143. doi: 10.3934/biophy.2018.2.125.

  1. It is known that DNA polymerases proceed 3'-5'- exonuclease activity with double-stranded DNA, digesting overhangs or protrusive ends until the start point of the double helix. How can be explained that the polymerases described in the work cleave single-stranded DNA (i.e., they function like exonuclease I)? The explanation have to be given.
  2. It is known that the Fe2+ ion is oxidized in air to Fe3+, which can not act as a cofactor for DNA polymerases. How was the oxidation of Fe2+ to Fe3+ excluded?

It would be useful to provide data on the specificity and fidelity of the polymerases as well as on their tolerance in regard to amplification inhibitors (for example, phenolic compounds, urea, hemin, etc.). However, this recommendation is optional.

English needs to be edited. Some words are used incorrectly.

In general, after the revisions, the article should be accepted for publication.

Author Response

Response to Reviewer

We sincerely thank you for your valuable and insightful comments that have significantly helped improve the quality of our manuscript. We have carefully addressed all the points, providing point-by-point responses in the revised file. The sections marked in green in both the revised manuscript and this response letter indicate the content that has been modified or newly added in response to the comments.

The authors have developed a new DNA polymerases with strand-displacement activity for isothermal amplification of nucleic acids. Various amplification techniques are widely used in many applications. It is important to develop enzymes that provide specific and highly sensitive detection of DNA or RNA. Thus, the problem that the authors dealt with is very relevant.

The manuscript includes data on the engineering and some properties of two novel DNA polymerases from Bacillus phage SRT01hs and Bacillus phage 3 BeachBum. The work seems very interesting and useful. However, to improve the quality of the manuscript, the authors should provide additional information. For this, the following issues need to be solved:

  1. In Introduction section, when describing isothermal amplification techniques, strand-displacement DNA polymerases and metal cofactors, it will be useful to cite some recent studies, e.g., the follows:

1) Sakhabutdinova AR, Garafutdinov RR. Mechanism of DNA multimerization caused by strand-displacement DNA polymerases. Anal Biochem. 2025 Aug;703:115876. doi: 10.1016/j.ab.2025.115876. 
2) Garafutdinov RR, Gilvanov AR, Kupova OY, Sakhabutdinova AR. Effect of metal ions on isothermal amplification with Bst exo- DNA polymerase. Int J Biol Macromol. 2020 Oct 15;161:1447-1455. doi: 10.1016/j.ijbiomac.2020.08.028. 

3) Vashishtha AK, Konigsberg WH. The effect of different divalent cations on the kinetics and fidelity of Bacillus stearothermophilus DNA polymerase. AIMS Biophys. 2018;5(2):125-143. doi: 10.3934/biophy.2018.2.125.

Response:

We thank the reviewer for this valuable suggestion. In the revised manuscript, we have integrated the recommended references within the Introduction and Discussion sections to contextualize our findings within broader literature.

“Mn2+ for Bst exo- polymerases has been shown as the most effective alternative cofactor to prevent multimerization which is non-specifical DNA synthesis [16, 20].”

“Interestingly, our results showed that Ca²⁺ failed to support polymerase activity of B family DNA polymerase Phi29, SRHS, and BBum. But for strand displacing A family DNA polymerases Bst, Ca²⁺ can still catalyze its extension [16]. This reveals the difference in cofactor requirements between distinct DNA polymerase families.”

[13] A.K. Vashishtha, W.H. Konigsberg, The effect of different divalent cations on the kinetics and fidelity of Bacillus stearothermophilus DNA polymerase, AIMS Biophys 5(2) (2018) 125-143.

[16] R.R. Garafutdinov, A.R. Gilvanov, O.Y. Kupova, A.R. Sakhabutdinova, Effect of metal ions on isothermal amplification with Bst exo- DNA polymerase, Int J Biol Macromol 161 (2020) 1447-1455.

[20] A.R. Sakhabutdinova, R.R. Garafutdinov, Mechanism of DNA multimerization caused by strand-displacement DNA polymerases, Anal Biochem 703 (2025) 115876.

  1. It is known that DNA polymerases proceed 3'-5'- exonuclease activity with double-stranded DNA, digesting overhangs or protrusive ends until the start point of the double helix. How can be explained that the polymerases described in the work cleave single-stranded DNA (i.e., they function like exonuclease I)? The explanation has to be given.

Response:

We sincerely appreciate this insightful question regarding the observed ssDNA degradation activity. In response, we have expanded our explanation in the revised manuscript. Our results demonstrated that both SRHS and BBum polymerases exhibited detectable 3'→5' exonuclease activity on ssDNA substrates. While classical proofreading exonucleases predominantly target double-stranded DNA, we proposed two non-exclusive mechanisms enabling ssDNA degradation:

(1) Structural flexibility in the exonuclease active site accommodating transiently folded ssDNA conformations; and

(2) Kinetic partitioning whereby the absence of polymerization shifts catalytic equilibrium toward exonuclease activity.

DNA polymerases inherently balance these competing activities. Under our assay conditions – which provide Mg²⁺ (essential for catalysis) but lack dNTPs and templates – the absence of polymerization establishes a pronounced kinetic bias favoring exonuclease function.

We have added a dedicated paragraph to the Discussion section.

“Although the exonuclease activity typically targeted double-stranded DNA (dsDNA), our results demonstrated that Phi29, SRHS, and BBum efficiently degraded single-stranded oligonucleotides (25bp). This ssDNA degradation capability—previously employed to confirm exonuclease activity of IME199 DNA polymerases [5]—differed from dedicated exonucleases like Exonuclease I (Exo I). Exo I is a non-processive exonuclease highly specific for ssDNA, whereas SRHS/BBum/Phi29/IME199 are processive enzymes whose exonuclease domain evolved primarily for dsDNA proofreading. We proposed two non-exclusive mechanisms enabling this ssDNA degradation: (1) structural flexibility of the exonuclease site, which accommodated transiently folded ssDNA; (2) kinetic partitioning, where the absence of polymerization shifted the equilibrium toward exonuclease activity [44]. Polymerases naturally balanced exonuclease and polymerase activities. However, under our exonuclease assay conditions—which pro-vided Mg²⁺ (essential for catalysis) but lacked dNTP substrates and templates—the absence of polymerization created a strong kinetic bias towards exonuclease activity.”

  1. It is known that the Fe2+ ion is oxidized in air to Fe3+, which can not act as a cofactor for DNA polymerases. How was the oxidation of Fe2+ to Fe3+ excluded?

Response:

Thank you for raising this critical technical issue.

In our Fe²⁺-dependent polymerase assays, we minimized Fe²⁺ oxidation by preparing all reactions using freshly prepared FeSO₄ under nitrogen-purged conditions. Additionally, we included a reducing agent, 1 mM TCEP.

We detailed nitrogen-purged conditions in the Materials and Methods section.

“To prevent the oxidation of Fe²⁺ to Fe³⁺ during RCA, all Fe²⁺-containing reactions were conducted under nitrogen-purged conditions. Specifically, FeSO₄ stock solutions were freshly prepared in deoxygenated water and kept on ice. Reaction buffers and enzyme mixtures were purged with ultrapure nitrogen gas for at least 30 min prior to use. Fe²⁺ was then added immediately before initiating the reactions.”

  1. It would be useful to provide data on the specificity and fidelity of the polymerases as well as on their tolerance in regard to amplification inhibitors (for example, phenolic compounds, urea, hemin, etc.). However, this recommendation is optional.

Response:

Thank you for this critical comment. We have added the fidelity data for the BBum_Pro_L and Phi29 DNA polymerases described in Supplementary File 1 and tested the tolerance for urea (Supplementary Figure 3).

“Fidelity assessment revealed BBum_Pro_L maintains accuracy comparable to Phi29 polymerase in the presence of Mg2+ [42], as determined by our novel isothermal amplification fidelity assay (Supplementary File 1). However, Mn2+ as metal cofactors resulted in lower fidelity than Mg2+(Supplementary File 1).”

“Besides, urea tolerance was assessed using RCA. As was shown in Supplementary Figure 3, urea inhibited extension efficiency of DNA polymerases in a concentration-dependent manner. Notably, at 1 M urea, BBum retained a certain level of amplification activity, outperforming both SRHS and Phi29 under the same conditions.”

English needs to be edited. Some words are used incorrectly.

Response:

Thank you for your suggestion. We have asked a native English speaker to help us correct the languages.

In general, after the revisions, the article should be accepted for publication.

Reviewer 3 Report

Comments and Suggestions for Authors

In this study, Yaping Sun et al identified and modified two B-family DNA pols from Bacillus phages SRT01hs and BeachBum, which can be used for the incorporation of non-natural nucleotides. These new enzymes have efficient incorporation activities under Mg²⁺ conditions and bringing new opportunities in the field of biotechnology. The research methods of the paper are rigorous, the data are detailed, the conclusions are reliable, and it has high academic value and application prospects. I think this paper is worthy of publication in Biomecules. 

I only recommend the aurthor to revise their figures, there are too much gels picture with mutiple lanes. It would be better to combine some results into one panel with columns or lines, which would be more readable.

Author Response

Response to Reviewer

We sincerely thank you for your valuable and insightful comments that have significantly helped improve the quality of our manuscript. We have carefully addressed all the points, providing point-by-point responses in the revised file. The sections marked in green in both the revised manuscript and this response letter indicate the content that has been modified or newly added in response to the comments.

In this study, Yaping Sun et al identified and modified two B-family DNA pols from Bacillus phages SRT01hs and BeachBum, which can be used for the incorporation of non-natural nucleotides. These new enzymes have efficient incorporation activities under Mg²⁺ conditions and bringing new opportunities in the field of biotechnology. The research methods of the paper are rigorous, the data are detailed, the conclusions are reliable, and it has high academic value and application prospects. I think this paper is worthy of publication in Biomecules. 

I only recommend the author to revise their figures, there are too much gels picture with mutiple lanes. It would be better to combine some results into one panel with columns or lines, which would be more readable.

Response

Thank you for your positive evaluation of our manuscript and your recommendation for publication in Biomolecules. We greatly appreciate your recognition of the novelty and potential applications of our engineered B-family DNA polymerases.

Regarding your valuable suggestion about the figures:

We agree that some of the gel images with multiple lanes may be visually complex. To address this, the revised manuscript features reorganized and streamlined versions of original Figures 1 and 4, where key quantitative results were now presented in column graphs. Representative gel images have been relocated to Supplementary Figure 2 and Supplementary Figure 7 for enhanced clarity.

Round 2

Reviewer 1 Report

Comments and Suggestions for Authors

Authors improve the manuscript  in accordance with comments.  

Author Response

Authors improve the manuscript in accordance with comments.

Response:We sincerely thank the reviewer for these insightful comments. We appreciate the opportunity to improve the clarity and accuracy of our work.